# The Influence of Iron-Deficiency Anaemia (IDA) Therapy on Clinical Outcomes and Healthcare Resource Consumptions in Chronic Kidney Disease Patients Affected by IDA: A Real-Word Evidence Study among the Italian Population

**DOI:** 10.3390/jcm11195820

**Published:** 2022-09-30

**Authors:** Valentina Perrone, Chiara Veronesi, Melania Dovizio, Domenica Daniela Ancona, Fausto Bartolini, Fulvio Ferrante, Alessandro Lupi, Stefano Palcic, Davide Re, Annamaria Pia Terlizzi, Antonio Ramirez de Arellano Serna, Paolo Cogliati, Luca Degli Esposti

**Affiliations:** 1CliCon Società Benefit S.r.l. Health, Economics & Outcomes Research, 40137 Bologna, Italy; 2Dipartimento Farmaceutico, Azienda Sanitaria Locale delle province di Barletta- Andria- Trani (BAT), 76125 Andria, Italy; 3Dipartimento Farmaceutico, Unità Sanitaria Locale Umbria 2, 05100 Terni, Italy; 4Dipartimento della Diagnostica ed Assistenza Farmaceutica, ASL Frosinone, 03100 Frosinone, Italy; 5Azienda Sanitaria Locale del Verbano Cusio Ossola (VCO), 28887 Omegna, Italy; 6Azienda Sanitaria Universitaria Integrata Giuliano-Isontina (ASUGI), 34148 Trieste, Italy; 7Dipartimento Assistenza Territoriale, Azienda Sanitaria Locale di Teramo, 64100 Teramo, Italy; 8CSL Vifor Group, 8152 Glattbrugg, Switzerland; 9CSL Vifor Italia, 00142 Rome, Italy

**Keywords:** direct healthcare costs, non-dialysis-dependent chronic kidney disease patients (ND-CKD), iron-deficiency anaemia (IDA), IDA therapy, real-world data analysis

## Abstract

Anaemia is a uraemia-related complication frequently found in non-dialysis-dependent chronic kidney disease (ND-CKD) patients, with iron-deficiency anaemia (IDA) as the main underlying mechanism. Given the suboptimal anaemia management in ND-CKD patients with a co-diagnosis of IDA, this study evaluated the role of IDA therapy on clinical outcomes and healthcare resource consumptions in an Italian clinical setting. A retrospective observational real-world analysis was performed on administrative databases of healthcare entities, covering around 6.9 million health-assisted individuals. From January 2010 to March 2019, ND-CKD patients were included and diagnosed with IDA in the presence of two low-haemoglobin (Hb) measurements. Patients were divided into IDA-treated and untreated, based on the prescription of iron [Anatomical-Therapeutic Chemical (ATC) code B03A] or anti-anaemia preparations (ATC code B03X), and evaluated during a 6-month follow-up from the index date [first low haemoglobin (Hb) detection]. IDA treatment resulted in significantly decreased incidence of all cause-related, cardiovascular-related, and IDA-related hospitalizations (treated vs. untreated: 44.5% vs. 81.8%, 12.3% vs. 25.3%, and 16.2% vs. 26.2%, respectively, *p* < 0.001). A healthcare direct cost estimation showed that overall mean expenditure per patient reduced by 47% with IDA treatment (5245€ vs. 9918€, *p* < 0.001), mainly attributable to hospitalizations (3767€ vs. 8486€, *p* < 0.001). This real-life analysis on Italian ND-CKD-IDA patients indicates that IDA therapy administration provides significant benefits in terms of patients’ clinical outcomes and healthcare cost savings.

## 1. Introduction

Anaemia commonly occurs as a complication in chronic kidney disease (CKD) patients [1,2] and has been associated with a reduced quality of life [3,4], a worse renal survival [5], an increase in morbidity and mortality [6,7], and higher healthcare costs [8]. 

Although it is known that the incidence and prevalence of anaemia rises with kidney function decline, this complication is also frequent in non-dialysis-dependent chronic kidney disease (ND-CKD) patients [9,10]. 

The pathogenesis of anaemia in CKD is multifactorial, but the impaired erythropoietin production and iron deficiency (iron-deficiency anaemia, IDA) are the primary mechanisms involved [11,12].

These pathophysiological processes are the basis for the current management of CKD-related anaemia, where erythropoiesis stimulating agents (ESAs) and iron supplementation represent the mainstays of treatment for ND-CKD patients [11,13]. 

While in patients under chronic hemodialysis, the benefits of high-dose I.V. iron regimens in terms of reduction of cardiovascular risk and deaths, transfusions, and ESA doses have been demonstrated [14], in the pre-dialysis phase, the optimal scheme for anaemia management, including treatment initiation, medication choice, and target biochemical levels to avoid possible adverse effects, remains controversial [15,16,17]. The results from clinical trials, indicating higher cardiovascular mortality associated with haemoglobin (Hb) levels higher than 11 g/dL in patients under erythropoietin stimulating agent (ESA) treatment, led to a definition of black box warning labels of ESA by the United States Food and Drug Administration [18].

In real-world practice, anaemia is suboptimally managed among ND-CKD patients, with a substantial proportion of them with low Hb levels and/or with IDA who remain untreated [19,20]. The elevated prevalence and unsatisfying management of anaemia in ND-CKD patients can be also ascribed to a persistent clinical inertia to iron supplementation and, to a lesser extent, but anyhow relevant, to ESA therapy [21]. Untreated anaemia has been associated with a reduced quality of life and increased cardiovascular disease, hospitalizations, cognitive impairment, and mortality [22]. There is evidence from international studies on ND-CKD Stages 3–5 patients that a low transferrin saturation (TSAT) index and ferritin have detrimental effects on health-related quality of life (HRQoL), further pointing out the need for interventional studies on iron therapy in this clinical setting [23].

The economic burden for the management of CKD is considerable [24,25], and a significant direct contributor to such high expenditures is CKD-related anaemia [26].

A recent meta-analysis showed that, in CKD anaemic patients, the non-treatment of CKD-related anaemia with ESA is associated with higher medical costs and health resource consumptions, highlighting the importance of treating this condition, as well as in view of its economic burden [26]. In addition, a recent literature review confirmed that the lack of treatment of anaemia in ND-CKD patients resulted in increased healthcare costs and poorer HRQoL [27], emphasizing the need for further investigations in this still open clinical challenge.

Thus, the present study aimed to evaluate, in a real-world setting in Italy, the characteristics of ND-CKD patients with a co-diagnosis of IDA and the impact of IDA therapy on clinical outcomes and healthcare resource consumptions.

## 2. Materials and Methods

### 2.1. Data Source

In this retrospective observational study, data were collected from Italian Healthcare Departments’ administrative databases, covering around 6.9 million health-assisted individuals. The following databases were used: (i) demographic database, which consists of all patient demographic data, such as gender, age, and death; (ii) pharmaceutical database, which supplies information on medicinal products reimbursed by the National Health Service (NHS) such as the Anatomical-Therapeutic Chemical (ATC) code, number of packages, number of units per package, unit cost per package, and prescription date; (iii) hospitalization database, which includes all hospitalization data for patients under analysis, such as the discharge diagnosis codes classified according to the International Classification of Diseases, Ninth Revision, Clinical Modification (ICD-9-CM), Diagnosis-Related Group (DRG), and DRG charge (provided by the Health System); (iv) outpatient specialist services database, which incorporates all information about visits and diagnostic tests (date and type of prescription, prescription activity, and laboratory test or specialist visit charge); and (v) payment exemption database, which contains data of the exemption codes that allow to avoid the contribution charge for services/treatments when specific diseases are diagnosed. 

An anonymous univocal numeric code was assigned to each study subject to guarantee patients’ data privacy, in full conformity with the European General Data Protection Regulation (GDPR) (2016/679). The patient’s code in each database permitted the electronic linkage between all databases. All of the results coming from the analyses were produced as aggregated summaries, and never attributable to a single institution, department, doctor, individual, or individual prescribing behaviours. The study has been presented to and approved by the local Ethics Committees of the Healthcare Departments involved in the study. 

### 2.2. Study Design and Study Population

Non-dialysis patients with CKD stage 3a/3b/4/5 diagnosis [defined according to an estimated glomerular filtration rate (eGFR) corresponding to 60–45 mL/min/1.73 m^2^, 44–30 mL/min/1.73 m^2^, 29–15 mL/min/1.73 m^2^, and < 15 mL/min/1.73 m^2^, respectively] or with CKD stage 3, 4, or 5 hospitalization [ICD-9-CM codes 585.3, 585.4, and 585.5 (primary or secondary diagnosis), respectively], across the period 01/2010–03/2019, were included. Enrolled ND-CKD patients were defined as with IDA co-diagnosis if presented with two low-Hb measurements (Hb < 13 g/dL (males); Hb < 12 g/dL (females), accordingly to Kidney Disease Improving Global Outcomes (KDIGO) clinical practice guidelines) [1], even not consecutive, during the enrolment period, considering the following criteria: (i) at least 1 week between the two low Hb measurements and (ii) no more than three months between the two Hb measurements. The date of the first low Hb measurement was considered as the index date for ND-CKD-IDA patients. In the absence of Hb measurement, the index date was the date of the first prescription of iron preparations (ATC code B03A) or other anti-anaemic preparations (ATC code B03X) (as proxy of diagnosis), or the date of the first hospitalization discharge date for IDA (ICD-9-CM code 280), as the primary or secondary diagnosis. IDA therapy was evaluated during 6 months of follow-up. The characterization and the follow-up periods lasted from the index date 1 year before and 6 months after, respectively. Patients were sub-grouped into IDA-treated and untreated (based on the prescription of iron preparations (ATC code B03A) or other anti-anaemic preparations (ATC code B03X)). The study design is detailed in Figure 1.

### 2.3. Analysis of Baseline Clinical and Demographic Characteristics of Patients

For all patients included in the study, baseline characteristics, namely age and gender, were evaluated. The comorbidities were identified from discharge diagnosis at primary and secondary levels. When a diagnosis was not available, the prescriptions of specific drugs were used as a proxy to determine the specific comorbidity. The following comorbidities were evaluated: cardiovascular diseases (by hospitalization, ICD-9-CM codes 410, 411, 413, 414, 430, 431, 432, 433, 434, 435, 436, 437, 438, 440, and 443); polycystic kidney disease (by hospitalization, ICD-9-CM code 753.1); autoimmune disease (by hospitalization ICD-9-CM codes 714, 720, 696, 555, and 556; exemption codes 006, 054, 045, and 009; or the prescription of immunosuppressant (ATC code L04A)); diabetic nephropathy (by hospitalization, ICD-9-CM code 250.4); chronic obstructive pulmonary disease (COPD) (by the prescription of drugs for obstructive airway diseases, ATC code R03 in patients aged > 40); infections (by hospitalization, ATC code J01); diabetes (by the prescription of antidiabetic agents, ATC code A10); and hypertension (by the prescription of antihypertensive medications, ATC codes C02, C03, C07, C08, and C09).

### 2.4. Evaluation of Outcomes

During the 6 months of follow-up from the index date, the occurrence of clinical outcomes and the time to outcome occurrence (in days), in terms of death, cardiovascular events (evaluated as reported above), tumor (identified by hospitalization, ICD-9-CM codes 140-239), and end-stage renal disease (ESRD) (identified by hospitalization, ICD-9-CM code 585.6), were recorded.

### 2.5. Healthcare Resource Consumption and Cost Analysis

In alive patients treated or not with IDA therapy, the healthcare resource utilization during 6 months of follow-up was evaluated in terms of the following variables: (i) mean number of drug prescriptions in the hospital setting; (ii) mean number of hospitalizations, in terms of all-cause hospitalizations, cardiovascular-related hospitalizations, and IDA-related hospitalizations [i.e., for anaemia (ICD-9-CM code 280), blood transfusions (ICD-9-CM codes V582, procedure 99.0)]; (iii) iron infusions (ICD-9-CM procedure code 99.2); and (iv) mean number of prescriptions for outpatients’ specialist services (OSS) reimbursed by the NHS. The direct healthcare costs were evaluated over the follow-up period and were related to the following resource consumptions: hospitalizations (determined using the DRG tariffs), drug costs (evaluated for those drugs reimbursed by the Italian NHS using its purchase price), and the OSS costs according to regional tariffs. Data were reported as the mean annual healthcare cost per patient.

### 2.6. Statistical Analysis

Continuous variables are reported as mean ± standard deviation (SD), while categorical variables are expressed as numbers and percentages. IDA-ND-CKD patients were categorized into the two cohorts, that is, patients treated with or without IDA therapy. The results were compared between the two cohorts and statistical significance was accepted at *p* < 0.05. Thus, the propensity score matching (PSM) methodology was applied to abate potential unbalances in baseline characteristics among the two cohorts (IDA-ND-CKD patients treated with or without IDA therapy). Patients were matched on quintiles of propensity score calculated using a logistic regression model, which includes all baseline characteristics listed above. To maintain the maximum number of patients, a 1:1 matching algorithm was used, i.e., for one patient without therapy, one with therapy was sampled. The standardized mean difference (SMD) was reported and values greater than 0.1 were considered for declaring an imbalance. All analyses were performed using Stata SE version 12.0 (StataCorp, College Station, TX, USA).

## 3. Results

During the entire inclusion period, 67,704 patients with ND-CKD-IDA diagnosis were enrolled, 57,802 (85.4%) in stage 3a/b, 7461 (11.0%) in stage 4, and 2441 (3.6%) in stage 5. Dividing the entire included patients by the presence/absence of IDA therapy, 31,631 (46.7%) were under treatment during 6 months after IDA identification and 36,073 (53.3%) were without IDA therapy (Figure 2).

After PSM, 29,782 ND-CKD-IDA patients with and without IDA therapy (mean age 79 years, 42% male), were included in the analysis of the study variables. The two cohorts were matched for all variables, except for the incidence of polycystic kidney disease, diabetic nephropathy, and hypertension (Table 1), which was slightly higher in the IDA-treated versus untreated cohort (polycystic kidney disease, 0.3% vs. 0.1%, respectively; diabetic nephropathy, 1.6% vs. 0.6%, respectively; and hypertension, 90.3% vs. 89%, respectively). In the three cases, absolute standard differences in disease incidence showed a value < 10%, which is considered as not significant [28].

During 6 months of follow-up, the occurrence of clinical outcomes and days to occurrence were evaluated (Table 2). Patients without IDA therapy compared with those with IDA therapy showed significantly higher rates of death (19.9% vs. 9.3%, respectively, *p* < 0.001), more frequent occurrence of cardiovascular events (12.1% vs. 5.4%, *p* < 0.001), and cancer (10.7% vs. 5.1%, *p* < 0.001). On the other hand, untreated versus treated patients had significantly lower number of days to death (69.0 ± 70.9 vs. 85.3 ± 69.7 days, *p* < 0.001), days to cardiovascular events (40.9 ± 42.4 vs. 54.7 ± 52.4, days, *p* < 0.001), and days to malignancy detection (43.4 ± 40.5 vs. 60.7 ± 49.6 days, *p* < 0.001). The rate of ESRD occurrence was slightly lower in the untreated cohort with respect to the IDA therapy group (2.1% vs. 2.7%, respectively, *p* < 0.001), and the time to ESRD was 49.1 ± 42.7 days for the untreated patients and 72.0 ± 50.4 days in those treated with IDA therapy (*p* < 0.001). 

In terms of healthcare resource use, during the six months of follow-up, in IDA-treated versus untreated alive patients, the mean number of drug prescriptions per patient was higher (13.4 ± 7.2 vs. 11.8 ± 7.1, *p* < 0.001), while the mean number of OSS prescriptions (7.2 ± 8.3 vs. 8.3 ± 8.9) and hospitalizations (0.8 ± 1.1 vs. 1.5 ± 1.2) was significantly lower (*p* < 0.001) (Table 3). 

A detailed investigation into hospitalizations showed that IDA-treated versus untreated patients had a significantly (*p* < 0.001) lower incidence of all cause-related (44.5% vs. 81.8%), cardiovascular-related (12.3% vs. 25.3%), and IDA-related hospitalizations (16.2% vs. 26.2%) (Figure 3). 

The estimation of healthcare direct costs covered by the Italian NHS revealed that the overall mean expenditure per patient was 5245€ for patients with IDA therapy and 9918€ for the untreated ones (*p* < 0.001), with a reduction of 47% in treated patients (Figure 4). In particular, this difference was principally attributable to hospitalization-related costs (IDA-treated vs. untreated: 3767€ vs. 8486€, *p* < 0.001), followed by OSS-related costs (529€ vs. 577€, *p* < 0.001), in front of a slight increase in drug-related costs (949€ vs. 855€, *p* < 0.01) (Figure 4).

## 4. Discussion

This real-world study investigated the effects of IDA therapy on the clinical and economic burden of CKD patients not yet under dialysis with a co-diagnosis of IDA, among the Italian population. Two cohorts of ND-CKD patients were compared, namely treated and untreated with IDA therapy, and the PSM methodology was applied to overcome potential selection bias in the retrospective analysis and to adjust covariates among the two cohorts. In the two covariate-adjusted cohorts, all baseline characteristics were superimposable, except for diabetic nephropathy, polycystic kidney disease, and hypertension frequencies, which were slightly higher (although with an absolute standard difference < 0.1) among IDA-treated patients.

In the present analysis, more than half of ND-CKD-IDA patients were untreated for the anaemic state. These data are consistent with previous reports by the Chronic Kidney Disease Outcomes and Practice Patterns Study (CKDopps) [29], where the prevalence of ESA prescriptions for individuals with Hb < 10 g/dL was 28% in the USA, 39% in Brazil, and 57% in Germany. Among patients with ferritin < 100 ng/mL who also had Hb < 12 g/dL, only 27–44% were prescribed with iron supplementation within 3 months after Hb measurement [29], thus indicating that iron replacement therapy remains considerably underestimated and a suboptimal management of CKD anaemic patients seems to be common in the clinical practice [21,29]. Given that the KDIGO guidelines indicate that iron or ESA treatment initiation is desirable in CKD adults with TSAT < 30% and ferritin < 500 ng/mL [1], our findings as well as other previously published data [21,29] seem to indicate a very cautious and conservative attitude in the decision of when to start therapy in the real-life clinical setting. Indeed, this hot point is further complicated by some divergences between KDIGO and CKDopps recommendations in haemoglobin values to define anemia and in thresholds of iron-related indices for initiating iron therapy [1,29]. Moreover, there might be variations in the conventional clinical practice between countries, substantiated by individualization of treatments using patient-tailored approaches and nephrologist preferences [1,29]. Recent Danish population-based data showed that increasing grades of anaemia in severe CKD patients are associated with worse clinical outcomes, including earlier dialysis initiation, major adverse cardiovascular events, acute hospitalization, and all-cause death [30]. These results confirmed those from several studies in selected populations indicating that untreated anaemia in CKD is strongly associated with cardiovascular complications [7,31,32,33,34,35] and subsequent increased hospitalizations and mortality. In this setting, studies addressing the effects of treating iron deficiency for improving patient outcomes are necessary. Despite the large amount of evidence proving that treating iron deficiency ameliorates cardiovascular risk, regardless of anaemia in patients with heart failure [7,34,35,36,37,38], data collected in the routine clinical practice of ND-CKD population are still inadequate [33].

Our findings demonstrated that patients under IDA treatments were characterized by a lower incidence of mortality and cardiovascular events occurred during the follow-up. Moreover, among ND-CKD-IDA patients under IDA therapy, a reduction in cancer-related hospitalizations was observed, in line with the recent findings reported in a population-based retrospective cohort study where an increased cancer risk in patients with IDA was reported [39]. On the other hand, it should be also underlined that ESAs are not recommended in oncological patients, in view of the very delicate balance between the benefits on anaemia and the increased risk of cancer mortality associated with these treatments [40]. Hence, in our sample population, the striking lower cancer incidence in the IDA-treated group (less than about an half) should not be viewed as direct effect of ESAs, also taking into account that 6 months of follow-up is a too short of a time to evaluate any kind of such causative association. It is rather possible that the lower numbers of tumor cases among patients with IDA treatment might reflect an “a priori” clinical decision. In a nutshell, given the risk of ESAs for cancer patients, there might have been a lower inclination by the clinicians to prescribe ESAs in the presence of malignancy. 

This analysis also highlighted a more frequent occurrence of ESRD in ND-CKD-IDA patients treated with IDA medications, but with a slow progression. One of the reasons for this clinical finding might lie in the higher percentage of IDA-treated patients with diabetic nephropathy, which is currently considered the leading cause of ESRD in high-income countries and likely worldwide [29,30,31]. 

Specifically, in patients under IDA therapy, a reduced occurrence of all-cause hospitalizations, as well as IDA-related and cardiovascular-related hospitalizations, was found, consistent with previous reports [24,33]. All these data are in line with studies that have shown that untreated anaemia in CKD is strictly associated with cardiovascular complications [34], resulting in increased hospitalizations [35] and mortality [41,42,43,44]. In patients under IDA medication, the total direct healthcare costs related to their management were significantly lower than those related to untreated IDA-ND-CKD patients, and this effect on cost-savings observed in patients under IDA therapy was mainly driven by the reduction in hospitalization expenses. A significant resource requirement is well recognized among patients with CKD, and several potentially modifiable factors, including the anaemic state, seem to be associated with increased resource utilization [44,45]. Thus, any effort to optimize anaemia management in CKD patients through IDA treatment can result in improved clinical outcomes and, ultimately, in substantially lessened healthcare expenditures [26,44,45,46].

The results must be interpreted in consideration of the limitations related to the study’s observational nature, based on data collected from administrative databases. Administrative databases have progressively improved the quality of the collected data, but some information may be missing; patients with missing data were excluded from the analysis. We were not able to include all clinical data and the occurrence of comorbidities were evaluated using a proxy of diagnosis. Hence, there might be an incomplete capture of these variables among patients and an underestimation of diagnoses; primary care data could not be collected by the administrative databases used here. In the cost and healthcare utilization analysis, a possible flaw could be that the databases captured only direct medical healthcare resource use reimbursed by the Italian NHS and we cannot account for indirect costs and those related to visits and expenses outside the national healthcare system. Moreover, in patients without ICD-9-CM codes for CKD, only one eGFR value was used to define the presence of CKD, whereas international guidelines require at least two pathological values or other markers of kidney damage more than 3 months apart [45]. This might have resulted in a partial overestimation of CKD cases. Besides, the evaluation of IDA treatments was based on prescriptions only, thus prescriptions could have been related to conditions other than anaemia of CKD. Patients from the two cohorts were subjected to PSM analysis by considering only baseline variables available in the analysis; thus, the impact of other variables has not been evaluated. Further limitations might lie in the relatively short follow-up and the variable duration of IDA therapy among treated patients. Lastly, although the present was a sponsored study, it has been conducted with the participation of several independent entities in compliance with all the principles to ensure data integrity, research honesty, and result objectivity.

In conclusion, the results of this real-life analysis among Italian ND-CKD-IDA patients showed that the administration of IDA therapy could positively impact clinical outcomes, in terms of mortality and occurrence of cardiovascular complications. Moreover, the treatment of anaemic state was associated with decreased healthcare resource use with a consequent reduction in direct costs covered by the Italian NHS, mainly related to hospitalizations. These findings suggest that the optimal management of the anaemic state in IDA-ND-CKD patients could be an important challenge in order to improve patients’ clinical outcomes. 

## Figures and Tables

**Figure 1 jcm-11-05820-f001:**
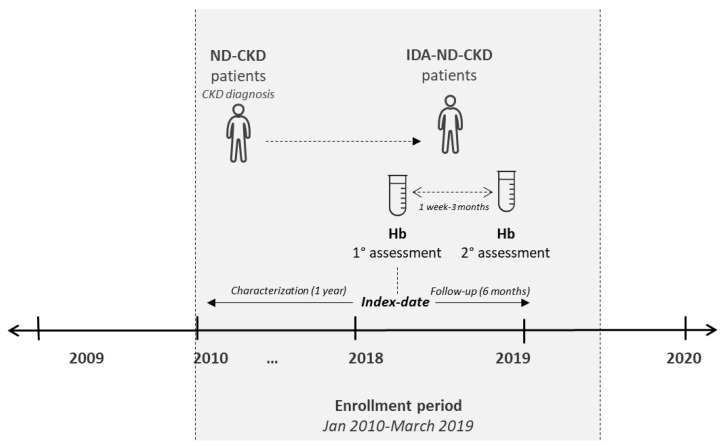
Schematic representation of the study design. Non-dialysis (ND) patients with a diagnosis of chronic kidney disease (CKD) (ND-CKD) were identified during the enrolment period; those presenting two low Hb measurements (with an interval between 1 week and 3 months) were defined as iron-deficiency anaemia (IDA)-ND-CKD.

**Figure 2 jcm-11-05820-f002:**
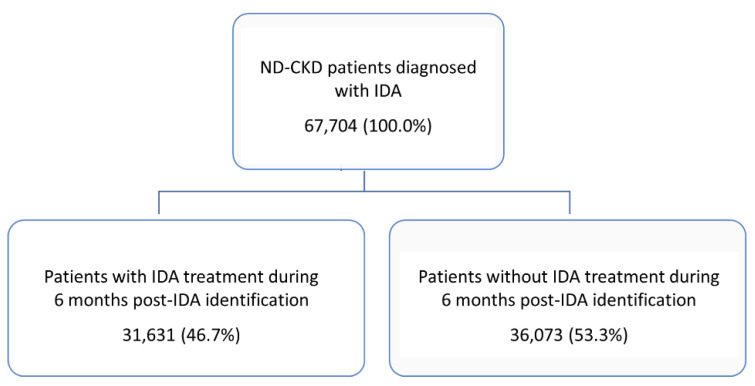
Identification of the target population and divided according to the presence/absence of iron-deficiency anaemia (IDA) therapy during the 6-month period after IDA identification. ND, non-dialysis; CKD, chronic kidney disease.

**Figure 3 jcm-11-05820-f003:**
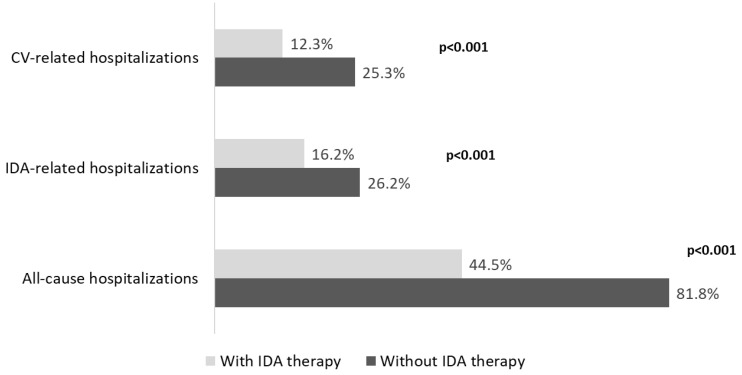
Analysis of all-cause and disease-related hospitalizations in ND-CKD-IDA patients treated or not with IDA therapy, during the follow-up, post PSM. CKD, chronic kidney disease; IDA, iron-deficiency anaemia; ND, non-dialysis; PSM, propensity score matching; CV, cardiovascular.

**Figure 4 jcm-11-05820-f004:**
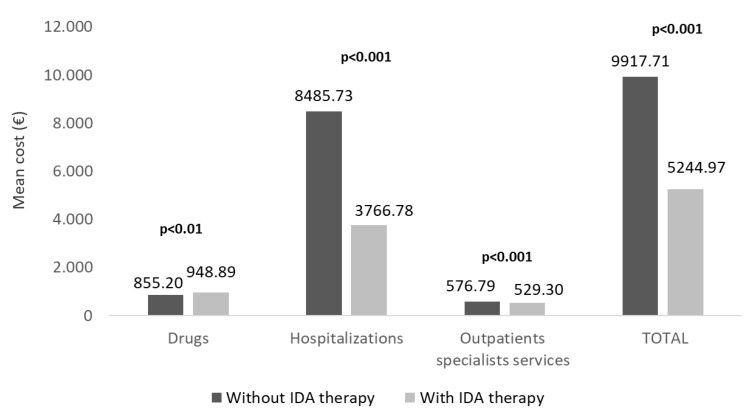
Mean total direct healthcare costs in alive ND-CKD-IDA patients treated or not with IDA therapy, during the follow-up, post PSM. CKD, chronic kidney disease; IDA, iron-deficiency anaemia; ND, non-dialysis; PSM, propensity score matching.

**Table 1 jcm-11-05820-t001:** Demographic and clinical characteristics of IDA-ND-CKD during the characterization period, by the presence of IDA therapy, post PSM. Significant *p*-values are highlighted in bold.

	Without IDA Therapy	With IDA Therapy	*p*-Value	SMD
*n*	29,782	29,782		
Age, mean ± SD	79.1 ± 11.1	79.0 ± 11.7	0.273	0.009
Male, *n* (%)	12,494 (42.0)	12,665 (42.5)	0.156	0.012
Cardiovascular disease, *n* (%)	3342 (11.2)	3431 (11.5)	0.251	0.009
Polycystic kidney disease, *n* (%)	26 (0.1)	79 (0.3)	**<0.001**	0.042
Autoimmune disease, *n* (%)	295 (1.0)	302 (1.0)	0.773	0.002
Diabetic nephropathy, *n* (%)	192 (0.6)	485 (1.6)	**<0.001**	0.093
COPD, *n* (%)	7340 (24.7)	7473 (25.1)	0.207	0.010
Infections, *n* (%)	18,566 (62.3)	18,657 (62.7)	0.441	0.006
Diabetes, *n* (%)	8880 (29.8)	9100 (30.6)	**0.050**	0.016
Hypertension, *n* (%)	26,496 (89.0)	26,902 (90.3)	**<0.001**	0.045

COPD, chronic obstructive pulmonary disease; CKD, chronic kidney disease; IDA, iron-deficiency anaemia; ND, non-dialysis; PSM, propensity score matching; SD, standard deviation; SMD, standardized mean difference.

**Table 2 jcm-11-05820-t002:** Clinical outcomes analysis in ND-CKD-IDA patients treated or not with IDA therapy, post PSM. Significant *p*-values are highlighted in bold.

	Without IDA Therapy	With IDA Therapy	*p*-Value
*n*	29,782	29,782	
Death, *n* (%)	5933 (19.9)	2756 (9.3)	**<0.001**
Cardiovascular events, *n* (%)	3589 (12.1)	1619 (5.4)	**<0.001**
Tumor, *n* (%)	3182 (10.7)	1506 (5.1)	**<0.001**
ESRD, *n* (%)	637 (2.1)	806 (2.7)	**<0.001**
Days to death, mean ± SD	69.0 ± 70.9	85.3 ± 69.7	**<0.001**
Days to cardiovascular events, mean ± SD	40.9 ± 42.4	54.7 ± 52.4	**<0.001**
Days to tumor, mean ± SD	43.4 ± 40.5	60.7 ± 49.6	**<0.001**
Days to ESRD, mean ± SD	49.1 ± 42.7	72.0 ± 50.4	**<0.001**

CKD, chronic kidney disease; ESRD, end stage renal disease; IDA, iron-deficiency anaemia; ND, non-dialysis; SD, standard deviation; PSM, propensity score matching.

**Table 3 jcm-11-05820-t003:** Healthcare resource use in ND-CKD-IDA alive patients treated or not with IDA therapy, during the follow-up, post PSM (data are given as mean ± SD per patient). Significant *p*-values are highlighted in bold.

	Without IDA Therapy	With IDA Therapy	*p*-Value
*n*	23,849	27,026	
Drugs, mean ± SD	11.8 ± 7.1	13.4 ± 7.2	**<0.001**
Hospitalizations, mean ± SD	1.5 ± 1.2	0.8 ± 1.1	**<0.001**
Outpatient specialist services, mean ± SD	8.3 ± 8.9	7.2 ± 8.3	**<0.001**

CKD, chronic kidney disease; IDA, iron-deficiency anaemia; ND, non-dialysis; PSM, propensity score matching; SD, standard deviation.

## Data Availability

All data used for the current study are available upon reasonable request to CliCon S.r.l. Società Benefit, which is the body entitled to data treatment and analysis by Local Health Units.

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
