# Peer review of "The Influence of Iron-Deficiency Anaemia (IDA) Therapy on Clinical Outcomes and Healthcare Resource Consumptions in Chronic Kidney Disease Patients Affected by IDA: A Real-Word Evidence Study among the Italian Population"

_jcm, 2022, doi:10.3390/jcm11195820_

Round 1

Reviewer 1 Report

Very interesting work. Anaemia, as the most common complication of CKD, is a great challenge for nephrologists. So far, more attention has been paid to anaemia in CKD 5D. This work proves not only the importance of anaemia in CKD 3-4, but also how many complications can be avoided by iron supplementation and at the same time reduce the financial cost of treating CKD patients.

Author Response

Comments and Suggestions for Authors

Very interesting work. Anaemia, as the most common complication of CKD, is a great challenge for nephrologists. So far, more attention has been paid to anaemia in CKD 5D. This work proves not only the importance of anaemia in CKD 3-4, but also how many complications can be avoided by iron supplementation and at the same time reduce the financial cost of treating CKD patients.

RE: We sincerely thank you for the positive feedback and for the time spent to revise our paper. The importance of timely treatment of anaemia to slow down the progression towards later CKD stages and ultimately to dialysis dependency has been also recently corroborated in a study by Gok MG and colleagues published in April 2022. The authors found that in both non-dialysis and hemodialysis patients, eryptosis, a mechanism of premature suicidal erythrocyte death, can be accelerated by inflammation and high PTH levels, but this process can be successfully counterbalanced by EPO and calcium blockers [Gok MG, et al, Int Urol Nephrol. 2023; doi: 10.1007/s11255-022-03207-3]. Consequently, the possibility to using targeted strategies to keep anaemia under control and to delay the progression towards chronic dialysis therapy in these patients can also imply positive rebounds on cost saving for the NHS, and we are glad that the Reviewer underlined this key message of our work.

Reviewer 2 Report

It is a very interesting manuscript that needs only some minor corrections. 

INTRODUCTION: Please, specify which are "anti-anemia preparations", this term is too inespecific and I suppose that you are talking about erythopoietin and similar drugs.

RESULTS: End of second paragraph: "

incidence showed a value <10% which is considered as not significant

Please, a reference is needed. 

"The rate of ESRD occurrence was slightly higher in patients under IDA-therapy with respect to the untreated cohort (2.7% vs. 2.1%, P<0.001), and the time to ESRD was 72. 0 ± 50.4 days for patients treated with IDA therapy and 49.1 ± 42.7 days for the untreated cohort. The rate of ESRD occurrence was slightly higher in patients under IDA-therapy with respect to the untreated cohort (2.7% vs. 2.1%, P<0.001), and the time to ESRD was 72. 0 ± 50.4 days for patients treated with IDA therapy and 49.1 ± 42.7 days for the untreated cohort" The beginning of the paragraph seems to be wrong, in table 2 ESRD incidence is lower in IDA-therapy group.

"The reduced total costs in IDA-treated patients were mainly driven by a reduction of hospitalization-related costs." Erase this part of the paragraph, the meaning is clear in the previous lines. 

DISCUSSION:

First paragraph: "In the two covariate-adjusted cohorts, all baseline characteristics were superimposable, except for diabetic nephropathy and hypertension frequencies." And also APKD, see table 1.

Second paragraph: "In the present analysis, more than half of ND-CKD-IDA patients were untreated for the anaemic state. These data are consistent with previous reports by the Chronic Kidney Disease Outcomes and Practice Patterns Study (CKDopps) [28], where the prevalence of ESA prescriptions for individuals with Hb <10 g/dL was 28% in the USA, 39% in Brazil, and 57% in Germany. Among patients with ferritin <100 ng/mL who also had Hb <12 g/dL, only 27–44% were prescribed with iron supplementation within 3 months after Hb measurement [28], thus indicating that iron replacement therapy remains considerably underestimated and a suboptimal management of CKD anaemic patients seems to be common in the clinical practice [21, 28]." This is a very conservative approach and real world seems to be worse, since KDIGO Guidelines recommend treatment when ferritin is <300.

Most important. The authors said that cancer incidence was lower in IDA treated patients. Since erithropoiesis stimulating agents are associated to high mortality in cancer patients, it would be very important to know how many people were receiving this class of drugs. On the other hand, may be that people with cancer were not receiving treatment due to this comorbidity, this should be discussed. 

Author Response

Comments and Suggestions for Authors

It is a very interesting manuscript that needs only some minor corrections.

We sincerely thank you for the overall positive feedback.

INTRODUCTION: Please, specify which are "anti-anemia preparations", this term is too inespecific and I suppose that you are talking about erythropoietin and similar drugs.

RE: We have added the ATC codes to every mention of iron preparations (ATC code B03A) or other antianaemic preparations (ATC code B03X), including the Abstract where they were missing (page 1, line 37). We did not detail specifically all the antianaemic preparation evaluated in the analysis, since the ATC code B03X univocally identifies the following: erythropoietin, darbepoetin alfa, methoxy polyethylene glycol-epoetin beta, peginesatide, toxadustat, luspatercept, daprodustat and vadadustat (https://www.whocc.no/atc_ddd_index/?code=B03XA&showdescription=no)

Incidence showed a value <10% which is considered as not significant

Please, a reference is needed.

RE: We have added the reference at n. 28 in the bibliography section, all the following ref. numbers have been shifted also in the text (https://www.ncbi.nlm.nih.gov/pmc/articles/PMC5761065/)

"The rate of ESRD occurrence was slightly higher in patients under IDA-therapy with respect to the untreated cohort (2.7% vs. 2.1%, P<0.001), and the time to ESRD was 72. 0 ± 50.4 days for patients treated with IDA therapy and 49.1 ± 42.7 days for the untreated cohort. The rate of ESRD occurrence was slightly higher in patients under IDA-therapy with respect to the untreated cohort (2.7% vs. 2.1%, P<0.001), and the time to ESRD was 72. 0 ± 50.4 days for patients treated with IDA therapy and 49.1 ± 42.7 days for the untreated cohort".  The beginning of the paragraph seems to be wrong, in table 2 ESRD incidence is lower in IDA-therapy group.

RE: This is an absolutely agreeable remark, and we now revised the entire paragraph to make it more understandable (page 6, lines 233-247 of the file open in review tracking modality). We followed the same order as in the table when describing comparisons, and so the untreated patients are always mentioned first and then the IDA-treated (as it is thorough the entire manuscript). In other words, consistent with all the other tables and figures in the manuscript, the comparisons are always described in the text as follows: IDA-untreated versus IDA-treated.

"The reduced total costs in IDA-treated patients were mainly driven by a reduction of hospitalization-related costs." Erase this part of the paragraph, the meaning is clear in the previous lines.

Re: Along with your reasonable suggestion, we have addressed this point in the amended version of the manuscript (page 8, lines 276-277).

DISCUSSION:

First paragraph: "In the two covariate-adjusted cohorts, all baseline characteristics were superimposable, except for diabetic nephropathy and hypertension frequencies." And also APKD, see table 1.

Re: This was a careless mistake, we apologize for that, and we have now corrected (page 8, line 290).

Second paragraph: "In the present analysis, more than half of ND-CKD-IDA patients were untreated for the anaemic state. These data are consistent with previous reports by the Chronic Kidney Disease Outcomes and Practice Patterns Study (CKDopps) [28], where the prevalence of ESA prescriptions for individuals with Hb <10 g/dL was 28% in the USA, 39% in Brazil, and 57% in Germany. Among patients with ferritin <100 ng/mL who also had Hb <12 g/dL, only 27–44% were prescribed with iron supplementation within 3 months after Hb measurement [28], thus indicating that iron replacement therapy remains considerably underestimated and a suboptimal management of CKD anaemic patients seems to be common in the clinical practice [21, 28]." This is a very conservative approach and real world seems to be worse, since KDIGO Guidelines recommend treatment when ferritin is <300.

Re: This is true, but also a tricky issue for clinicians, since the KDIGO and CKDopps recommendations have some divergence in describing haemoglobin thresholds to define anemia and in thresholds of iron-related indices for initiating iron therapy. In addition, there are variations in the real-life clinical practice between countries, substantiated by individualization of treatment often using patient-tailored approaches and nephrologist preferences. We have added some short hints about this point in the revised version (pages 8-9, lines 301-310).

Most important. The authors said that cancer incidence was lower in IDA treated patients. Since erythropoiesis stimulating agents are associated to high mortality in cancer patients, it would be very important to know how many people were receiving this class of drugs. On the other hand, may be that people with cancer were not receiving treatment due to this comorbidity, this should be discussed.

RE: Among our overall sample population of 67,704 ND-CKD-IDA patients, 31,631 (46.7%) were treated with IDA therapy during 6 months after IDA identification, and 36,073 (53.3%) were untreated. After PSM to balance the two populations, the comparisons were carried out between 29,782 for each group. So, as Table 2 shows, there were 3,182 (10.7%) with neoplasm among those without IDA treatment and the 1,506 (5.1%) in the treated ones. This lower incidence in the treated group (less than an about half) is highly significant and raises attention. Anyhow, although we are not able to provide the precise numbers of those patients with anemia and cancer for whom the clinicians decided to avoid ESAs, the well-known effects of these agents on cancer mortality might feasibly represent the reason for this striking difference [Bohlius J, et al, Lancet. 2009 doi:10.1016/S0140-6736(09)60502-X]. So, the lower numbers of cancer patients among the IDA-treated patient should not be viewed in terms of an effect of ESAs (anyhow 6 months of follow-up are a too short time to evaluate such an association), but rather as a consequence of an “a-priori” decision of the clinicians. In a nutshell, given the risk of ESAs for cancer patients, there might be a lower inclination by the clinicians to prescribe ESAs in the presence of malignancy. This was reported at page 9, lines 328-338.